# Family Processes, Parenting Practices, and Psychosocial Maturity of Chinese Youths: A Latent Variable Interaction and Mediation Analysis

**DOI:** 10.3390/ijerph18084357

**Published:** 2021-04-20

**Authors:** Jerf W. K. Yeung

**Affiliations:** Department of Social and Behavioural Sciences, City University of Hong Kong, Kowloon, Hong Kong; ssjerf@gmail.com

**Keywords:** family processes, parenting practices, psychosocial maturity, self-concept, self-control, perspective taking

## Abstract

Development of psychosocial maturity has profound implications for youths’ well-being and positive development in the long run. Nevertheless, little research has investigated the way family socialization contributes to youths’ psychosocial maturity. Both the concepts of family socialization and psychosocial maturity are multifaceted and latent, which may lead to biased results if studied by manifest variables. Also, no existing research has discovered how different family socialization components interact latently to contribute to youths’ psychosocial maturity. The current study, based on a sample of 533 Chinese parent-youth dyads, examined the effects of family socialization by positive family processes and authoritative parenting, and their latent interaction in an integrated moderation and mediation modeling framework on Chinese youths’ psychosocial maturity. Results showed that both positive family processes and authoritative parenting, and their latent interaction significantly predicted the higher psychosocial maturity of Chinese youths. Authoritative parenting acted as a mediator for the relationship between positive family processes and Chinese youths’ psychosocial maturity. Furthermore, the mediating effect of authoritative parenting was conditioned by different contexts of positive family processes, the strongest and least strong effects found in high and low positive family processes, respectively, and moderate effect observed in medium positive family processes. Findings of the current study contribute to our understanding of the complicated family mechanism in relation to youth development, especially in this digital era.

## 1. Introduction 

Youth development refers to various aspects of cognitive, psychological, and behavioral transformations, for which research has recently pointed to the importance of youths’ establishment of psychosocial maturity, due to its profound impact on youths’ long-term adjustment and well-being [1,2]. Psychosocial maturity is a cognitive construct, closely related to the psychological health and behavioral adaptation of youths. Moreover, although family is deemed the fundamental and most proximal socialization agent for youth development [3,4], research on the way family socialization shapes youth development of psychosocial maturity is limited, particularly in Chinese societies. Family has long been regarded as the most important societal and interpersonal unit affecting individual growth and development in Chinese culture [4,5]. Hence, it is important to scrutinize the relationship between family socialization and Chinese youths’ development of psychosocial maturity. Prior family research generally considered parenting behavior as tantamount to the concept of family socialization, which is indeed multifaceted and comprises family processes and parenting practices [4,6]. Therefore, more research is needed to discover how family processes and parenting practices, the two main family socialization components, concurrently contribute to youths’ development of psychosocial maturity.

Moreover, most existing studies have generally treated family socialization by family processes and parenting practices, and youths’ psychosocial maturity as observable variables [6,7,8], which induce a concern of methodological fallacy. Theoretically, family processes, parenting practices, and psychosocial maturity are conceptualized by multiple attributes and indicators that are expected to be coherently related and mutually reinforced [2,6,7,9]; therefore, they should be considered as latent constructs. If latent constructs are treated as manifest variables in the study relationships, biased and misleading findings may ensue [10,11]. What is more, although family processes and parenting practices represent distinguishable but mutually related facets of family socialization, the way their interaction leading to youths’ development of psychosocial maturity is uncharted. Some research has reported that family processes set a socialization context in regulating the effects of parenting practices on youth development [7,12]. Furthermore, parenting practices are empirically found to be heavily affected by the function of family processes [13,14], denoting a possible mediating role in the relationship between family processes and youths’ psychosocial maturity. In sum, this study intends to investigate the way family processes and parenting practices, and their interaction commonly contribute to the psychosocial maturity of Chinese youths. Parenting practices are believed to mediate the relationship between family processes and Chinese youths’ psychosocial maturity, and family processes would interact with parenting practices to predict Chinese youths’ psychosocial maturity by moderating the mediating effect of parenting practices on Chinese youths’ psychosocial maturity.

## 2. Theoretical Framework of the Current Study 

### 2.1. The Importance of Psychosocial Maturity to Youth Development

Consistently, youth development of psychosocial maturity has been empirically supported as an essential intrapersonal resource, influential of other aspects of psychological and behavioral health [1,2,8], such as enhanced academic and social performance and deceased substance use, delinquency, and emotional problems. However, little is known about the way family socialization by family processes and parenting practices contributes to youths’ psychosocial maturity. Although a consensus on defining psychosocial maturity is lacking, Steinberg and Cauffman [15] constructed a model of psychosocial maturity that includes temperance, responsibility, and perspective, to present the key intrapersonal elements of psychosocial maturity. Specifically, temperance indicates one’s ability of self-control and self-regulation. Responsibility refers to the assurance of self-worth and having a clear sense of self for avoidance of negative environmental and peer influences. Additionally, perspective means having the capability to take the viewpoint of others [2,8,16]. Accordingly, the psychosocial maturity of Chinese youths considered in this study is composed of the intrapersonal characteristics of self-control, positive self-concept, and consideration of others, which are believed to commonly converge on a latent construct of psychosocial maturity. Clearly, youths’ psychosocial maturity is closely related to their resilience and long-term well-being, corresponding to what Chassin, et al. [17] stated: “(p)sychosocially mature individuals take responsibility for their actions, autonomously rely on personal resources, and have a clear sense of identity. They exhibit temperance in curbing impulsive and aggressive behavior and are able to adopt multiple temporal and social perspectives” (pp. 48–49). This is resonant with the latent trait theory, suggesting that the general psychological responses, emotional expressions, and behavioral choices of individuals are the direct projection of their intrapersonal latent traits and dispositions [18]. Hence, a person of high self-control, better self-worth and identity, and perspective taking will be more effective in curbing external temptations and aggressiveness, valuing her or his social role and functions competently, and being more considerate of the needs of others [2,15], leading to better individual, interpersonal, and social betterment.

Simmons, Fine, Knowles, Frick, Steinberg, and Cauffman [16] found that the psychosocial maturity of justice-involved male youths not only significantly moderated the promotive effect of their callous-unemotional traits on delinquency during the year following their first arrest, but was also directly predictive of less reoffending. Monahan, Steinberg, Cauffman, and Mulvey [9] reported that different patterns of psychosocial maturity development significantly existed in a sample of serious juvenile offenders that predicted their different trajectories of antisocial behavior. Youths who persisted in antisocial behavior significantly demonstrated deficits in elements of psychosocial maturity, compared to their counterparts who desisted from antisocial behavior from adolescence to early adulthood. Recently, Yeung [2] corroborated the protective effects of youths’ psychosocial maturity on alleviating their internalizing and externalizing symptoms. Thus, as family has been deemed the fundamental and most influential socialization agent in Chinese societies, it is research worthy to scrutinize the relationships of family processes and parenting practices in connection to Chinese youths’ psychosocial maturity.

### 2.2. Family Socialization and Psychosocial Maturity

The family provides the most influential and intimate socialization experiences for youths to develop their cognitive, psychological, and behavioral competence [4,19,20]. Less clear is how family processes and parenting practices concurrently contribute to youths’ psychosocial maturity [2,6]. Generally, family processes refer to the overall family climate, relational interactions, mutual support, interpersonal cohesion, and communications among family members, especially for parents, in the family realm [13,21]. Parenting practices denote direct parental guidance, instructions, rules, and standards set by parents for raising their children in the expected direction and life orientation they hope for [5,6]. However, extant family research conducted in Chinese societies has predominantly focused on the way parenting behavior affects youth development [22,23], leaving the influence of family processes less examined. As family plays the crucial role of socialization and individual welfare in Chinese culture [4,24], both family processes and parenting practices should be considered equally important when investigating the relationship between family socialization and Chinese youths’ psychosocial maturity. This approach is valid, as family processes highlight the importance of informal socialization by modeling and inculcating family relationships, communication, and interactional dynamics, and parenting practices denote the influence of formal social control by learning and fulfilling parental instructions, requirements, and standards [7,21]. Both correspond to the emphasis of Chinese culture on interpersonal cohesion, attachment, and respect for authority [24,25].

Limited research exists on exploring how the way family socialization by family processes and parenting behavior shapes youth development of psychosocial maturity, including in Chinese societies. In their earliest work, Steinberg, et al. [26] reported that authoritative parenting was significantly predictive of adolescent psychosocial maturity and academic achievement. Adolescent psychosocial maturity acted as a mediator for the association between authoritative parenting and academic achievement. Further, Beckmeyer and Russell [3] found that family management practices by parental knowledge, behavioral control, parental academic involvement, and unsupervised time with peers were significantly related to youths’ psychosocial maturity across different family structures. Yeung [2] confirmed that both positive family processes and authoritative parenting significantly predicted youths’ psychosocial maturity, which in turn led to their lower internalizing and externalizing problems. Indeed, social cognitive theory posits that one’s proximal experiences and contextual influences, such as family socialization, may greatly formulate her or his cognitive orientation, psychosocial expectation, and attitudinal selection by establishing apposite judgments, rationality, and values [27,28]. Therefore, both positive family processes and authoritative parenting in this study are expected to create a socialization context that promotes Chinese youths’ psychosocial maturity by cultivating their positive self-concept, behavioral regulation, and ability of perspective taking.

Although family processes and parenting practices are two distinguishable family socialization components, the latter is expected to be substantially swayed by the former [4,20]. The reason is that positive family processes imply the family realm filling with harmony, supportive relationships, mutual understanding, and cohesive interactions among family members, especially for parents. This may facilitate family social capital and resources for parents to perform effective parenting practices. The reverse is true for a disorganized home environment and conflicting family relationships that may adversely hamper parenting quality. Yeung and Chan [13] found that family processes significantly predicted authoritative parenting, both of which contributed to the mental health of their young children. Likewise, Valiente, Lemery-Chalfant, and Reiser [14] reported that a chaotic home climate and discordant family interactions directly contributed to poor parenting practices, due to fatigue and tension imposed on parents by suboptimal family processes. Therefore, authoritative parenting is expected to be a function of positive family processes to mediate the effect of positive family processes on Chinese youths’ psychosocial maturity.

### 2.3. Latent Nature of Studying Family Socialization and Psychosocial Maturity

Family socialization by positive family processes and authoritative parenting, and youths’ psychological maturity refer to the social concepts that are not directly observable, multifaceted, and need to be measured by multiple indicators [2,6,8]. Accordingly, they should be considered as unobserved entities to indicate their latent nature, by assigning meaning through the measurement of variances and covariances from their respective indicators, decomposing the covariance matrix into the constituent matrices of factor loadings, factor variances, and residual error terms [29]. The decomposition has to be expressed as a structural equation measurement model, to preclude unnecessary measurement errors, with the form $\sum^{\;}\left(\theta \right)={\Lambda \psi \Lambda }^{\prime }\;+⊛$, where $\sum^{\;}\left(\theta \right)$ is the variance-covariance matrix of observed indictors, Λ is the factor loading matrix, $\Lambda^{\prime }\;$means the transposed factor loading matrix, ψ is the factor variance matrix, and ⊛ connotes the measurement residual matrix. If researchers treat latent constructs instead with manifest variables, the variances of the study relationships are inflated and their standard errors then shrink [11,29], leading to biased and inaccurate results.

However, existing family studies have tended to examine the effects of family socialization by family processes and parenting practices on youth development, including psychosocial maturity, as manifest variables, which overlook their latent nature. In their earliest and pioneering study, Steinberg, Elmen, and Mounts [26] investigated the relationship between authoritative parenting and American youths’ psychosocial maturity by treating authoritative parenting and youth psychosocial maturity as independent manifest variables. They only separately analyzed the effects of parental acceptance, psychological autonomy, and parental control on the independent components of youths’ psychosocial maturity, such as self-reliance, work orientation, and identity, or their observable psychosocial maturity by combining the averages of self-reliance, work orientation, and identity. Similarly, Beckmeyer and Russell [3] examined the effects of family management practices according to parental knowledge, behavioral control, unsupervised time with peers, and academic involvement on youths’ psychosocial maturity measured by observable variables. Recently, Yeung [2] analyzed how family processes and parenting practices concurrently shape the latent construct of youths’ psychosocial maturity. However, both family processes and parenting practices were considered as manifest predictors and overlooked their possible latent interaction in relation to youths’ psychosocial maturity. Examining the latent interactional effect of family processes by parenting practices on youths’ psychosocial maturity is important, as recent research supports the moderating role of family processes in the connection between parenting and youth development [7,30]. In this study, positive family processes and authoritative parenting are considered latent constructs to predict Chinese youths’ psychosocial maturity. Positive family processes would interact with authoritative parenting latently, to contribute to better psychosocial maturity of Chinese youths. The strongest effect of authoritative parenting is expected to appear in the context of high positive family processes, and the least strong effect of authoritative parenting is expected to appear in the context of low positive family processes. The moderate effect of authoritative parenting is expected to appear in the context of medium positive family processes.

### 2.4. The Present Study

The current study takes a latent variable approach to investigate the effects of positive family processes and authoritative parenting, and their interaction on Chinese youths’ psychosocial maturity. Authoritative parenting is expected to mediate the effect of positive family processes on Chinese youths’ psychosocial maturity, and positive family processes would moderate the mediating effect of authoritative parenting on Chinese youths’ psychosocial maturity. Accordingly, the hypotheses are set below.

**Hypotheses** **1.**
*Both positive family processes and authoritative parenting and their latent interaction would positively predict Chinese youths’ psychosocial maturity.*


**Hypotheses** **2.**
*Authoritative parenting would mediate the relationship between positive family processes and Chinese youths’ psychosocial maturity.*


**Hypotheses** **3.**
*Positive family processes would moderate the mediating effect of authoritative parenting on Chinese youths’ psychosocial maturity, in which the strongest mediating effect is expected in high positive family processes, and the least strong mediating effect is expected in low positive family processes, and the moderate mediating effect is expected in medium positive family processes.*


The sociodemographic covariates of family composition, family welfare dependency, youth gender, and age were adjusted in the study relationships because prior research demonstrated their effects on youth development. Specifically, family composition refers to whether the youth participant lives with both biological mother and father or not (two biological parents family vs. other family). Family welfare dependency indicates whether the participating family receives any financial subsidies from the government. Youths living with both biological mother and father, referring to two biological parents family structure, and without welfare dependency would exhibit better development [4,20,31]. Being female and older would exhibit more maturity and considerateness [4,32]. In the modeling procedures, family composition, family welfare dependency, youth gender and age were adjusted as control variables to preclude their confounding effects.

## 3. Methods

### 3.1. Sample and Procedures

Data of the current study came from a community sample of 533 Chinese parent-youth dyads that were recruited with the help of service units in a large local NGO and Chinese churches. The main purpose of the study was to examine the way family socialization contributes to the psychological and behavioral domains of Chinese youth development in Hong Kong. Therefore, supervisors of service units in the NGO and religious leaders of churches were first contacted by the principal investigator of this study (the author), who explained the purpose and usefulness of the study to seek the support of locating eligible Chinese parent and youth participants. Consequently, 22 service units of the NGO and 43 local churches helped to locate eligible parent-youth dyads for the study. For selection criteria, the parent participant must be the biological mother or father and main caregiver of the youth participant in the home, to ensure knowledge of the development of the youth. The youth participants must be between 14 and 21 years old, which means in middle and late adolescence and early adulthood. These are the critical developmental stages for young people [30,33]. Specifically, if the participating family had more than one child within the targeted age range, the child who had just passed a birthday was selected. However, if there were two or more children in the same household eligible for the study, a twin for example, the one first born was selected. To ensure personal privacy, questionnaires were contained in independent envelops separately for the parent and child participants, and they returned their completed questionnaires in the same envelops. This procedure would increase variance of the parent-youth dyads by enhancing random selection and privacy. Participation in the current study was voluntary, and parental consent and youths’ assent were obtained before data collection. The study was ethically approved by the ethical review committee of City University of Hong Kong.

### 3.2. Measures

Positive family processes were measured by the 26-item Family Functioning Style Scale (FFSS) [34]. An example item is “We take pride in even the smallest accomplishments of our family members”. In this study, a multi-informant approach, rather than a single-rater perspective, was used to assess family processes, to increase reliability and objectivity. The ratings of parent and youth participants were combined to measure positive family processes, to reduce shared method variance bias [2,35]. As positive family processes were considered a latent construct and measured by the 26-item FFSS, which raises the concern of model parsimony and non-normality, item-parceling was used to construct the latent construct of positive family processes by creating three parcels through randomly assigning items 1, 4, 7, 10, 13, 16, 19, 22, and 25 to parcel one; items 2, 5, 8, 11, 14, 17, 20, 23, and 26 to parcel two; and items 3, 6, 9, 12, 15, 18, 21, and 24 to parcel three, to reduce possible correlated errors and enhance better-fitting solutions. Random assignment of items to form parcels is preferable to using exploratory factor analysis for parceling [36,37], as the latter fits less well with the assumption of measurement unidimensionality, imbedded in latent constructs. Composite reliability of positive family processes was excellent, ρ_c_ = 0.971, with mode fit CFI = 1.000 and RMSEA = 0.000.

Authoritative parenting was measured by the 10-item Authoritative Parenting subscale of the Parental Authority Questionnaire (PAQ) [38]. An example item is “My mother tells me how we should act and explains to us the reasons why”. A multi-informant approach was used to assess authoritative parenting by combing parent and youth participants’ ratings. The original scale was designed for children, so the items were rephrased to allow responses of parent participants. A modified example item is: “I tell my children how they should act and explain to them the reasons why”. Modification of measurement for a specific research need is common in empirical studies [33,35]. Item-parceling was also employed to reduce the number of indicators. Three parcels were created by randomly assigning items 1, 4, 7, and 10 to parcel one; items 2, 5, and 8 to parcel two; and items 3, 6, and 9 to parcel three. Composite reliability was excellent, ρ_c_ = 0.920, with model fit, CFI = 1.000 and RMSEA = 0.000.

Psychosocial maturity is composed of self-concept, self-control, and perspective taking, commonly used to converge on a latent construct of youths’ psychosocial maturity. Youth self-concept was measured by the 6-item Positive Self-image Scale [39], which was developed to measure the “positive self” in a sample of representative youths. It has good internal reliability [33,39]. An example item is “You have a lot of good qualities”. Youth self-control was measured by the 7-item Good Self-Control Scale [40], which has been used to assess youths’ self-regulation and persistence and has good internal consistency [12,40]. An example item is: “I stick with what I’m doing until I’m finished with it”. Further, youth perspective taking was measured by the 7-item Consideration of Others subscale of Weinberger Adjustment Inventory [41], which is commonly used to evaluate the perspective taking of others and has good internal reliability [1,2]. An example item is: “I try very hard not to hurt other people’s feelings”. Composite reliability was adequate, ρ_c_ = 0.692, with model fit, CFI = 1.000 and RMSEA = 0.000.

Sociodemographic covariates of family composition, family welfare dependency, youth gender, and age were adjusted in the study relationships. Family composition (1 = two biological parents family, 0 = otherwise), family welfare dependency (1 = welfare dependency, 0 = otherwise), and youth gender (1 = female, 0 = male) are dummy variables, and youth age is a count variable in exact years.

### 3.3. Analytic Procedures

Latent variables structural modeling was used to analyze the effects of positive family processes and authoritative parenting and their latent interaction on Chinese youths’ psychosocial maturity, which can be expressed as
(1)η=Βη+Γξ+ζ
where η is the latent outcome and endogenous variables, and ξ refers to exogenous latent variables that are connected in a system of linear equations by the beta coefficient matrices Β and gamma Γ, and a residual term of zeta ζ. To estimate the latent interaction of positive family processes x authoritative parenting and the mediation of authoritative parenting in relation to Chinese youths’ psychosocial maturity, the latent moderated structural equations approach (LMS) was applied in an integrated moderation and mediation modeling framework [42,43]. Specifically, the form of general interaction model of latent variables is written as
(2)$$\eta =\alpha +\gamma_{1}\xi_{1}+\gamma_{2}\xi_{2}+\gamma_{3}\xi_{1}\xi_{2}+\zeta$$
where α is the intercept, $\gamma_{1}$ and $\gamma_{2}$ are the first order effects, and $\gamma_{3}$ represents the latent interaction effect by the product term of $\xi_{1}\xi_{2}$, and an integrated moderation and mediation modeling framework means conducting interaction and mediational analyses synchronously in the same modeling procedure. Using LMS conducting interaction of latent variables has an advantage over existing conventional methods, e.g., constrained product indicator analysis (CPI), including its accounting for measurement errors, avoidance of nonlinear constraints for model identification, and less susceptibility to the multivariate normality assumption [43]. Maximum likelihood estimation with robust standard errors (MLR) was used to estimate the modeling procedures, due to its better capability in integrating data non-normality and observation non-independence [36,42]. The Monte Carlo method was employed for numerical integration because its usefulness for higher-dimensional integrals is superior to deterministic approaches [36]. All modeling procedures were conducted by Mplus 8.4 [44].

## 4. Results

Table 1 presents the sociodemographic characteristics of the parent-youth dyads: 88.4% came from two biological parents family, and 11.6% were from other family structure; and 43.9% of the participating families were welfare-dependent, and 56.1% were non-dependent. The gender of the main caregiver parents was 80.1% mothers and 19.9% fathers. Additionally, 57.4% of the youth participants were female and 42.6% were male. Their average age was 16.30, meaning generally in middle adolescence.

Table 2 shows the correlation coefficients of the main study variables, in which positive family processes were substantially correlated with authoritative parenting, r = 0.744, *p* < 0.001. Moreover, positive family processes were significantly correlated with youth self-concept, self-control, and perspective taking, r = 0.323, 0.263, and 0.280, *p* < 0.001. Authoritative parenting was significantly correlated with youth self-concept, self-control, and perspective taking, r *=* 0.242, 0.283, and 0.315, *p* < 0.001. Besides, youth self-concept, self-control, and perspective taking were significantly and concretely correlated with each other, ranging from r *=* 0.363 to 0.481, *p <* 0.001.

The first structural equation model was to set positive family processes and authoritative parenting as two latent constructs to predict the latent psychosocial maturity of Chinese youths while controlling for family composition, family welfare dependency, youth gender, and age concomitantly. The structural model had a good model-data fit: CFI *=* 0.978, RMSEA *=* 0.054, X^2^
*=* 144.140, df *=* 56, X^2^/df *=* 2.579. However, the modification index indicated regressing positive family processes on family composition and authoritative parenting on youth age. A better-fit structural model emerged, CFI = 0.982, RMSEA *=* 0.050, X^2^ = 126.204, df *=* 54, X^2^/df = 2.337. Figure 1 shows the standardized effects of positive family processes and authoritative parenting on Chinese youths’ psychosocial maturity (model 1). Specifically, both positive family processes and authoritative parenting significantly predicted the higher psychosocial maturity of Chinese youths, β = 0.185 and 0.326, *p* < 0.05 and 0.001. And positive family processes significantly and robustly contributed to more authoritative parenting, β = 0.790, *p* < 0.001. Older youths significantly presented better psychosocial maturity, β = 0.109, *p* < 0.05, and were negatively related to authoritative parenting, β *=* −0.089, *p* < 0.05. Additionally, parent-youth dyads from two biological parents family background significantly exhibited higher positive family processes compared to their counterparts of other family structure, β *=* 0.133, *p* < 0.01. Furthermore, the indirect effect test corroborated that authoritative parenting significantly mediated the relationship between positive family processes and Chinese youths’ psychosocial maturity (Table 3), β_ind_ = 0.257, *p* < 0.001.

Another structural model was constructed to test whether positive family processes and authoritative parenting and their latent interaction significantly predict Chinese youths’ psychosocial maturity. The structural model was then constrained to vindicate the moderated mediating effects of authoritative parenting on Chinese youths’ psychosocial maturity conditioned by positive family processes. Figure 2 shows that the latent interaction of positive family processes and authoritative parenting was significantly predictive of higher Chinese youths’ psychosocial maturity (model 2), β_mz_ = 0.084, *p <* 0.05. In addition, positive family processes and authoritative parenting still significantly predicted higher Chinese youths’ psychosocial maturity, β *=* 0.191 and 0.287, *p* < 0.05 and 0.001, and positive family processes significantly contributed to better authoritative parenting, β = 0.791, *p* < 0.001. Two biological parents family structure significantly exhibited better positive family processes when compared with other family structure, β = 0.124, *p* < 0.05, and older Chinese youths significantly had better psychosocial maturity, β = 0.108, *p* < 0.05. Model constraint was applied to set the latent predictor of positive family processes into low, medium, and high levels. First, the latent predictor of positive family processes was fixed to variance = 1 and classified into low, medium, and high levels by setting at 1 standard deviation below and above its latent mean structure. Then, the mediation of authoritative parenting was tested in the relationship between positive family processes and Chinese youths’ psychosocial maturity, conditioned by the three contextual levels set (model codes in Appendix A). The mediating effect of authoritative parenting was strongest at high positive family processes (Table 3), β_mz_ = 0.268, *p* < 0.001, and least strong at low positive family processes, β_mz_
*=* 0.208, *p* < 0.01, plus moderate mediating effect of authoritative parenting at medium positive family processes found, β_mz_
*=* 0.238, *p* < 0.01. This shows that the contribution of authoritative parenting to Chinese youths’ psychosocial maturity was significantly conditioned and varied by family processes. Lastly, lower Akaike’s information criterion (AIC) and sample-size adjusted Bayesian information criterion (BIC_c_) values of model 2 denote its better fit in predicting Chinese youths’ psychosocial maturity, AIC = 9490.267 and BIC_c_
*=* 9531.053, when compared to model 1, AIC *=* 9493.674, and BIC_c_ = 9533.358.

## 5. Discussion

Existing family studies have generally considered family socialization as a manifest entity [6,7,30]. Most youth research has measured youths’ psychosocial maturity as an observable outcome [3,9,16], which is in fact latent and may generate biased results if this methodological concern remains unresolved. One of the merits of the current study is the construction of a latent structural model to investigate the way positive family processes and authoritative parenting concurrently contribute to Chinese youths’ psychosocial maturity. Results show that both positive family processes and authoritative parenting significantly predict higher Chinese youths’ psychosocial maturity. In addition, the structural model with latent interaction by positive family processes and authoritative parenting appears to have a better model fit in predicting Chinese youths’ psychosocial maturity. This is evidenced by its lower AIC and BIC_c_ values and the significant interactional effect, explicating that not only family processes and parenting practices, but also their interaction, collectively constitute the core of family socialization contributing to youth development of psychosocial maturity. However, due to the cross-sectional design of the current study, future research should take a transitional approach to study the way changes in one facet of family socialization, e.g., family processes, may cause fluctuations in other family socialization components that interactively and longitudinally affect youth development.

A notable point of this study is that the effect of authoritative parenting on youths’ psychosocial maturity was stronger than that of positive family processes. For example, their respective effects in model 1 are β *=* 0.326 vs. 0.185, *p* < 0.001 and 0.05, and in model 2 are β = 0.287 vs. 0.191, *p* < 0.001 and 0.05. These different effects may evince the different socialization features of family processes and parenting practices in forming youth development. Family processes have more vicarious functions of socialization for youth development through the cultivation of the home climate and interpersonal interactions [4,21]. Parenting practices refer to more direct standards and requirements for youth development set by parental guidance and instructions reactively [6,7]. Therefore, it is valid to observe a more substantial influence of authoritative parenting on Chinese youths’ psychosocial maturity that needs to be trained up by close caregivers for how to manage self-image and values, perform self-regulation and persistence, and consider the needs of others [2,3]. Nevertheless, family processes and parenting practices may contribute to youth outcomes differently, depending on the outcome traits [4,20]. Comparatively, family processes are expected to be more influential on youths’ emotional and psychological development, such as subjective happiness and life satisfaction. Parenting practices are thought to more directly contribute to youths’ cognitive and behavioral growth, such as psychosocial maturity and decision making. Therefore, much research is required to scrutinize and clarify the different impacts of family processes and parenting practices on different aspects of youth development. Besides, as aforementioned, family socialization is a multifaceted concept, in which parental beliefs and parent-youth attachment as well as emotional expressions of family members are all influential of youth development [45,46,47]. However, we currently know little regarding how these cognitive and relational dimensions of family socialization work with different parenting styles and family processes to shape youth development [4,6,48]. Thus, to take an integrative approach to investigate the effects of different family socialization facets and their interactive effects on youth development concomitantly is important to enhance our understanding the relationship between family socialization and youth development.

Apparently, both positive family processes and authoritative parenting are important family socialization components contributing to better development of Chinese youths’ psychosocial maturity. This signifies that informal home atmosphere by daily interactions among family members, exhibition of mutual support, and expression of appreciation and concern and formal parental guidance by standards, rules, moral values, and life orientations commonly have inextricable influences on the development of Chinese youths’ psychosocial maturity. However, adolescence is a critical developmental period that implies the existence of unexpected difficulties and obstacles to harm youths’ development of psychosocial maturity [9,49]. Hence, more research is needed to scrutinize the effects of intrapersonal and environmental challenges of youths in interplay with family socialization on youth development [4,17,50]. Further, we still have not confirmed whether family processes and parenting practices present comparable significant effects on other aspects of youth development [4,19], such as substance use and emotional difficulties. Authoritative parenting was found as a function of positive family processes, which are then predictive of Chinese youths’ psychosocial maturity. Therefore, it is plausible that parenting behavior acts as a crucial mediator to transit the effects of family relationships, the couple’s cohesion, marital intimacy, and other home interpersonal dynamics on youth development [13,51]. Thus, future longitudinal research can help answer the inquiry by comparing the effects of family processes and parenting practices on various youth outcomes and clarifying the mediating role of parenting behavior in the relationship between family socialization and youth development.

Although both family processes and parenting practices influence Chinese youths’ psychosocial maturity, the former evidently moderates the effect of the latter, explicating that family contextual influences created by family processes heavily sway and regulate the effect of parenting practices on youth development [12,14]. Hence, it is reasonable to scrutinize whether other contextual conditions of parents, e.g., their working relationships and employment environment, may exert equal moderating effects on their parenting behavior, which in turn shapes youth development. As such, it important to verify the susceptibility of parenting practices by different contextual influences and validate its buffering or magnifying function for youth development in different contextual conditions. This is especially important for youth development in the current digital era, which characterizes the inundation and overflow of multifarious information and cultural influences from multiple contextual sources through the internet and online devices, possibly leading youths away from positive development [48,52]. Therefore, better understanding of the way of family processes and parenting practices, and their latent interaction contributing to youths’ psychosocial maturity concurrently, can more effectively help parents and their offspring do better in this information age.

## 6. Conclusions

In sum, the current study corroborated that family socialization is a multifaceted concept that latently and interactively contributed to youth development in a dynamic way, for which both family processes and parenting practices and their latent interaction are found synchronously influential on Chinese youths’ psychosocial maturity. Thereby, it is suggested that researchers, family practitioners, educators, and social workers should take a multitudinous perspective to consider and scrutinize the impacts of different facets of family socialization on youth development, scrupulously.

Lastly, the current study contains some limitations that must be addressed in future research. First, the nonrandom sample of parent-youth dyads makes the generalizability of the study findings difficult; and the cross-sectional design of the study precludes the causality of the structural relationship between family socialization by positive family processes and authoritative parenting and Chinese youths’ psychosocial maturity. Second, this study only examined authoritative parenting in relation to youths’ psychosocial maturity, leaving the influences of other parenting styles and their interaction with family processes on Chinese youth development unknown. In fact, authoritarian parenting was found to bring some benefits to youths in non-Western cultures [53,54], which reveals the importance of investigating and comparing different parenting styles in relation to youth development under the regulation of family processes across different cultural contexts. Third, youth development involves cognitive, psychological, emotional, and behavioral dimensions, concurrently, that are mutually reinforced and related. Therefore, psychosocial maturity refers to only one facet of youth development, which leaves the whole picture of youth development uncharted. Fourth, measurement invariance of the study variables across different cultural and social contexts is important to ensure external validity of the proposed theoretical structures and findings of the current study [55,56]. Therefore, cross-cultural research is suggested in the future to vindicate the tenability of the influences of family socialization on youths’ psychosocial maturity, as cultural factors profoundly influence the patterns of family socialization and youth development [4,48]. Fifth, except family socialization, youth development is concomitantly susceptible to the effects of many social systems, such as school, peer network, and neighborhood. Accordingly, if future research employs a longitudinal design based on representative data to investigate how family socialization, in opposition to the influences of other social systems, shapes various aspects of youth development contemporaneously, family researchers can know more about the dynamic role that family plays in youth life.

## Figures and Tables

**Figure 1 ijerph-18-04357-f001:**
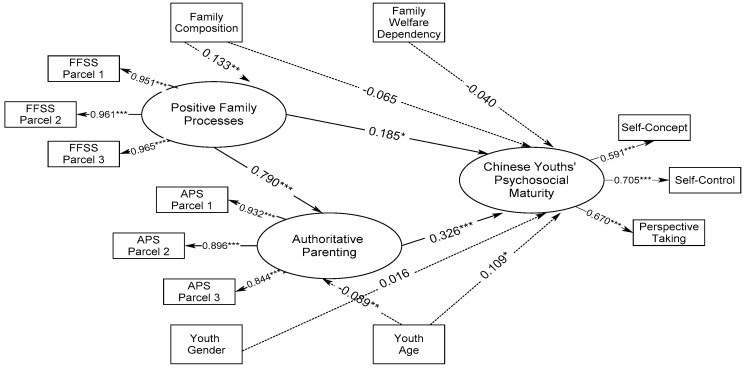
Structural Relationships of Positive Family Processes and Authoritative Parenting in Prediction of Psychosocial Maturity of Chinese Youths (Model 1). Note: Model fit is *CFI* = 0.982, *RMSEA* = 0.050, *X^2^* = 126.204, *df* = 54, *X^2^/df* = 2.337, *AIC* = 9493.674, and *BIC_c_* = 9533.358. * *p* < 0.05; ** *p* < 0.01; *** *p* < 0.001.

**Figure 2 ijerph-18-04357-f002:**
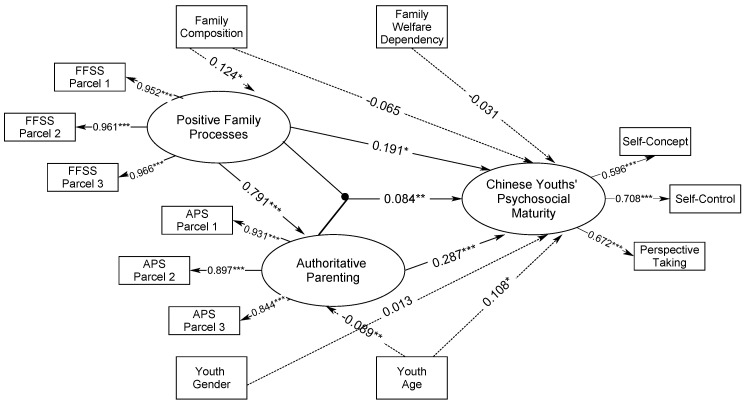
Structural Relationships of Positive Family Processes and Authoritative Parenting and Their Latent Interaction in Prediction of Psychosocial Maturity of Chinese Youths (Model 2). Note. *AIC* = 9490.267, and *BIC_c_ =* 9531.053. ** p* < 0.05; ** *p* < 0.01; *** *p* < 0.001.

**Table 1 ijerph-18-04357-t001:** Descriptive Statistics of Sociodemographic Characteristics of the Parent-Youth Participants.

	Mean/Frequency	SD/%
Family Composition ^a^		
Two Biological Parents Family	471	88.40%
Other Family	62	11.60%
Family Welfare Dependency		
Dependency	234	43.9%
Non-Dependency	299	56.10%
Parent Gender		
Female	427	80.1%
Male	106	19.9%
Youth Gender		
Female	306	57.40%
Male	227	42.60%
Youth Age	16.30	1.97

Note. ^a^ Two biological parents family refers to youth participants living with two biological parents of opposite gender, and other family is otherwise.

**Table 2 ijerph-18-04357-t002:** Adjusted Correlations of Positive Family Processes, Authoritative Parenting, Youth Self-Concept, Self-Control, and Perspective Taking.

		1.	2.	3.	4.	5.
1.	Positive Family Processes	--				
2.	Authoritative Parenting	0.744 ***				
3.	Youth Self-Concept,	0.323 ***	0.242 ***			
4.	Youth Self-Control,	0.263 ***	0.283 ***	0.425 ***		
5.	Youth Perspective Taking	0.280 ***	0.315 ***	0.363 ***	0.481 ***	--

*** *p*< 0.001.

**Table 3 ijerph-18-04357-t003:** Mediation of Authoritative Parenting and Moderated Mediation by Positive Family Processes and Authoritative Parenting in Prediction of Chinese Youths’ Psychosocial Maturity.

**Mediating Effect**	**β_ind_**	**SE**	**Z-Value**	**95% CI**
Authoritative Parenting	0.257	0.072	3.582 ***	0.116 to 0.398
**Moderated Mediating Effect**	**β_mz_**	**SE**	**Z-Value**	**95% CI**
Low Positive Family Processes	0.208	0.076	2.731 **	0.059 to 0.357
Medium Positive Family Processes	0.238	0.073	3.257 **	0.095 to 0.381
High Positive Family Processes	0.268	0.072	3.733 ***	0.127 to 0.409

** *p* < 0.01; *** *p* < 0.001.

## Data Availability

The data presented in this study are available on request from the corresponding author.

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
