# Peer review of "Family Processes, Parenting Practices, and Psychosocial Maturity of Chinese Youths: A Latent Variable Interaction and Mediation Analysis"

_ijerph, 2021, doi:10.3390/ijerph18084357_

Round 1

Reviewer 1 Report

The manuscript analyzes the relationship between authoritative parenting, family process and psychosocial maturity (self-concept, self-control and perspective taking) in China. Participants were 533 Chinese parent-youth dyads. It seems that both parent and his/her provide adolescent child family measures (parenting and family process) and it was used the mean. Overall, the results showed that family functioning and authoritative parenting have a positive impact on adolescent psychosocial maturity: adolescents tend to develop greater self-concept, self-control and perspective taking.

The introduction requires including within the text of manuscript important previous and recent main findings of the literature.

# Authors only examine one type of parenting (authoritative) and its impact on psychosocial maturity. The variable of parenting (effective parenting practices) should be labeled as authoritative parenting.

Authoritative parenting is commonly defined in parenting studies as the combined use of responsiveness and demandingness (Darling & Steinberg, 1993; Garcia, Serra, Zacares, Calafat, & Garcia, 2020). Studies with European-American middle-class families reveal the benefits of authoritative parenting (Lamborn et al., 1991), but authoritarian parenting (responsiveness without demandingness) is related to some benefits among ethnic minorities (e.g, Chinese-American, Chao, 2001). Overall, the influence of parenting could not be always the same in all cultural contexts (Pinquart & Kauser, 2018; Queiroz et al., 2020). The impact of parenting in Western Societies (Fuentes et al., 2019) could not be the same as in Eastern societies (Dwairy & Achoui, 2006), so parenting research need to test the influence of parenting on adolescent competence across different cultural contexts.

# Psychosocial maturity represent the capacity to function effectively on one’s own (i.e., individual adequacy), to interact adequately with others (i.e., interpersonal adequacy), and to contribute to social cohesion (i.e., social adequacy). Psychosocial maturity is a goal of parental socialization, but also should be considered as an outcome of parental socialization. Therefore, different indicators of psychosocial maturity (Garcia & Serra, 2019; Greenberg et al., 1975). Additionally, it should be included the idea that adolescence is a developmental time related to some vulnerability and difficulties, in part due to not all adolescents have enough psychosocial maturity (Fuentes et al., 2020; Greenberg et al., 1975).

#Sample description: More details about sample description are required. For parents: percentage of mothers/fathers Mean of age? Standard deviation of age?

#It is required a table of correlations between the main variables of the study. The table should be included in the text.

#Additionally, in the discussion, it is required that authors consider the importance of the invariance of the measures by sex, age and country. In order to compare the main findings from empirical studies across the world, it is necessary to find out whether the measures of the constructs are culturally invariant, in other words, whether the meaning of the items is the same for respondents in different cultural contexts (Chen et al. 2020; Garcia et al., 2018).

#As limitation it is required include that only one type of parenting (authoritative) is examined.

References

Chao, R. K. (2001). Extending research on the consequences of parenting style for Chinese Americans and European Americans. Child Development, 72, 1832-1843.   doi:10.1111/1467-8624.00381

Chen, F., Garcia, O. F., Fuentes, M. C., Garcia-Ros, R., & Garcia, F. (2020). Self-Concept in China: Validation of the Chinese version of the Five-Factor Self-Concept AF5 Questionnaire. Symmetry, 12(798), 1-13.   doi:10.3390/sym12050798

Darling, N., & Steinberg, L. (1993). Parenting style as context: An integrative model. Psychological Bulletin, 113, 487-496. doi:10.1037/0033-2909.113.3.487

Dwairy, M., & Achoui, M. (2006). Introduction to three cross-regional research studies on parenting styles, individuation, and mental health in Arab societies. Journal of Cross-Cultural Psychology, 37, 221-229.   doi:10.1177/0022022106286921

Fuentes, M. C., Garcia, O. F., & Garcia, F. (2020). Protective and risk factors for adolescent substance use in Spain: Self-esteem and other indicators of personal well-being and ill-being. Sustainability, 12(5967), 1-17. doi:10.3390/su12155962

Fuentes, M. C., García-Ros, R., Pérez-González, F., & Sancerni, D. (2019). Effects of parenting styles on self-regulated learning and academic stress in Spanish adolescents. International Journal of Environmental Research and Public Health, 16(2778), 1-19.

Garcia, F., Martínez, I., Balluerka, N., Cruise, E., García, O. F., & Serra, E. (2018). Validation of the Five-Factor Self-Concept Questionnaire AF5 in Brazil: Testing factor structure and measurement invariance across language (Brazilian and Spanish), gender and age. Frontiers in Psychology, 9(2250), 1-14.   doi:10.3389/fpsyg.2018.02250

Garcia, O. F., & Serra, E. (2019). Raising children with poor school performance: Parenting styles and short- and long-term consequences for adolescent and adult development. International Journal of Environmental Research and Public Health, 16(1089), 1-24.   doi:10.3390/ijerph16071089

Garcia, O. F., Serra, E., Zacares, J. J., Calafat, A., & Garcia, F. (2020). Alcohol use and abuse and motivations for drinking and non-drinking among Spanish adolescents: Do we know enough when we know parenting style? Psychology and Health, 35, 645-664. doi:10.1080/08870446.2019.1675660

Greenberger, E., Josselson, R., Knerr, C., & Knerr, B. (1975). The measurement and structure of psychosocial maturity. Journal of youth and Adolescence4(2), 127-143. doi:10.1007/BF01537437

Lamborn, S. D., Mounts, N. S., Steinberg, L., & Dornbusch, S. M. (1991). Patterns of competence and adjustment among adolescents from authoritative, authoritarian, indulgent, and neglectful families. Child Development, 62, 1049-1065.   doi:10.1111/j.1467-8624.1991.tb01588.x

Pinquart, M., & Kauser, R. (2018). Do the associations of parenting styles with behavior problems and academic achievement vary by culture? Results from a meta-analysis. Cultural Diversity and Ethnic Minority Psychology, 24, 75-100.   doi:10.1037/cdp0000149

Queiroz, P., Garcia, O. F., Garcia, F., Zacares, J. J., & Camino, C. (2020). Self and nature: Parental socialization, self-esteem, and environmental values in Spanish adolescents. International Journal of Environmental Research and Public Health, 17(3732), 1-13. doi:10.3390/ijerph17103732

Author Response

Thank you for reviewing the manuscript titled “Family Processes, Parenting Practices, and Psychosocial Maturity of Chinese Youths: A Latent Variable Interaction and Mediation Approach”, and now I have revised the manuscript according to the comments of the reviewers. My responses are below:

For Reviewer 1

# Authors only examine one type of parenting (authoritative) and its impact on psychosocial maturity. The variable of parenting (effective parenting practices) should be labeled as authoritative parenting.

Reply: Now effective parenting practices are all renamed to authoritative parenting.

# Authoritative parenting is commonly defined in parenting studies as the combined use of responsiveness and demandingness (Darling & Steinberg, 1993; Garcia, Serra, Zacares, Calafat, & Garcia, 2020). Studies with European-American middle-class families reveal the benefits of authoritative parenting (Lamborn et al., 1991), but authoritarian parenting (responsiveness without demandingness) is related to some benefits among ethnic minorities (e.g, Chinese-American, Chao, 2001). Overall, the influence of parenting could not be always the same in all cultural contexts (Pinquart & Kauser, 2018; Queiroz et al., 2020). The impact of parenting in Western Societies (Fuentes et al., 2019) could not be the same as in Eastern societies (Dwairy & Achoui, 2006), so parenting research need to test the influence of parenting on adolescent competence across different cultural contexts.

Reply: Agree, now the manuscript has incorporated the mentioning of cultural differences in sway of the effects of parenting styles on youth development, which is written:

“Second, this study only examined authoritative parenting in relation to youths’ psychosocial maturity, leaving the influences of other parenting styles and their interaction with family processes on Chinese youth development unknown. In fact, authoritarian parenting was found to bring some benefits to youths in non-Western cultures [48,49], which reveals the importance of investigating and comparing different parenting styles in relation to youth development under the regulation of family processes across different cultural contexts.” (Lines 2636-2642, Conclusion)

  1. Pinquart, M.; Kauser, R. Do the associations of parenting styles with behavior problems and academic achievement vary by culture? Results from a meta-analysis. Cult Divers Ethnic Minor Psychol 2018, 24, 75–100.
  2. Pinquart, M.; Gerke, D. C. Associations of parenting styles with self-esteem in children and adolescents: A meta-analysis. J Child Fam Stud 2019, 28, 2017–2035.

# Psychosocial maturity is a goal of parental socialization, but also should be considered as an outcome of parental socialization. Therefore, different indicators of psychosocial maturity (Garcia & Serra, 2019; Greenberg et al., 1975). Additionally, it should be included the idea that adolescence is a developmental time related to some vulnerability and difficulties, in part due to not all adolescents have enough psychosocial maturity (Fuentes et al., 2020; Greenberg et al., 1975).

Reply: Agree, the idea of adolescence as a transitional and challenging period for youth development has mentioned, which is written:

“However, adolescence is a critical developmental period that implies the existence of unexpected difficulties and obstacles to harm youths’ development of psychosocial maturity [9,44]. Hence, more research is needed to scrutinize the effects of intrapersonal and environmental challenges of youths in interplay with family socialization on youth development [4,17,45]. Further, we still have not confirmed whether family processes and parenting practices have comparable significant effects on other aspects of youth development [4,19], such as substance use and emotional difficulties.” (Lines 2382-2386, Discussion)

  1. Yeung, J. W. K. Religion, Family, and Chinese Youth Development. Routledge: London, UK, 2021.
  2. Monahan, K. C.; Steinberg, L.; Cauffman, E.; Mulvey, E. P. Trajectories of antisocial behavior and psychosocial maturity from adolescence to young adulthood. Dev Psychol 2009, 45, 1654–1668.
  3. Chassin, L.; Dmitrieva, J.; Modecki, K.; Steinberg, L.; Cauffman, E.; Piquero, A. R.; Knight, G. P.; Losoya, S. H. Does adolescent alcohol and marijuana use predict suppressed growth in psychosocial maturity among male juvenile offenders? Psychol Addic Behav 2010, 24, 48–60.
  4. Morawska, A. The effects of gendered parenting on child development outcomes: A systematic review. Clin Child Fam Psychol Rev 2020, 23, 553–576.
  5. Fuentes, M. C.; Garcia, O. F.; Garcia, F. Protective and risk factors for adolescent substance use in Spain: Self-esteem and other indicators of personal well-being and ill-being. Sustainability 2020, 12, 5962.
  6. Garcia, O. F.; Serra, E. Raising children with poor school performance: Parenting styles and short- and long-term consequences for adolescent and adult development. Int J Environ Res Public Health 2019, 16, 1089.

# Sample description: More details about sample description are required. For parents: percentage of mothers/fathers Mean of age? Standard deviation of age?

Reply: More information about sampling procedures and selection criteria of the study sample of Chinese parent-youth dyads are added, which are written:

“Data of the current study came from a community sample of 533 Chinese parent-youth dyads that were recruited with the help of service units in a large local NGO and Chinese churches. The main purpose of the study was to examine the way family socialization contributes to the psychological and behavioral domains of Chinese youth development in Hong Kong. Therefore, supervisors of service units in the NGO and religious leaders of churches were first contacted by the principal investigator of this study (the first author), who explained the purpose and usefulness of the study to seek the support of locating eligible Chinese parent and youth participants. Consequently, 22 service units of the NGO and 43 local churches helped to locate eligible parent-youth dyads for the study. For selection criteria, the parent participant must be the biological mother or father and main caregiver of the youth participant in the home, to ensure knowledge of the development of the youth. The youth participants must be between 14 and 21 years old, which means in middle and late adolescence and early adulthood. These are the critical developmental stages for young people [30,33]. Specifically, if the participating family had more than one child within the targeted age range, the child who had just passed a birthday was selected. However, if there were two or more children in the same household eligible for the study, a twin for example, the first born one was selected. To ensure personal privacy, questionnaires were contained in independent envelops separately for the parent and child participants, and they returned their completed questionnaires in the same envelops. This procedure would increase variance of the parent-youth dyads by enhancing random selection and privacy. Participation in the current study was voluntary, and parental consent and youths’ assent were obtained before data collection. The study was ethically approved by the ethical review committee of the City University of Hong Kong.”  (Lines 1132-1137, Methods)

  1. Everri, M.; Mancini, T.; Fruggeri, L. Family functioning, parental monitoring and adolescent familiar responsibility in middle and late adolescence. J Child Fam Stud 2015, 24, 3058–3066.
  2. Yeung, J. W. K. Parenting discrepancies in the aggregate parenting context and positive child outcomes in Chinese parent–child dyads. Pers Indiv Differ 2016, 98, 107–113.

# #It is required a table of correlations between the main variables of the study. The table should be included in the text.

Reply: Correlations of the study variables are added in the Results, which present:

“Table 2 shows the correlation coefficients of the main study variables of positive family processes, authoritative parenting, youth self-concept, self-control, and perspective taking. Consistently, positive family processes were substantially correlated with authoritative parenting, r=.744, p<.001. Moreover, positive family processes were significantly correlated with youth self-concept, self-control, and perspective taking, r= .323, .263, and .280, p< .001. Authoritative parenting was significantly correlated with youth self-concept, self-control, and perspective taking, r= .242, .283, and .315, p< .001. Youth self-concept, self-control, and perspective taking were significantly and concretely correlated with each other, ranging from r=.363 to .481, p< .001.” (Lines 1845-1853, Results)

Table 2. Adjusted Correlations of Positive Family Processes, Authoritative Parenting, Youth Self-Concept, Self-Control, and Perspective Taking.

1.

2.

3.

4.

5.

1.

Positive Family Processes

--

2.

Authoritative Parenting

.744***

3.

Youth Self-concept,

.323***

.242***

4.

Youth Self-control,

.263***

.283***

.425***

5.

Youth Perspective Taking

.280***

.315***

.363***

.481***

--

*p< .05; **p< .01; ***p< .001

# Additionally, in the discussion, it is required that authors consider the importance of the invariance of the measures by sex, age and country. In order to compare the main findings from empirical studies across the world, it is necessary to find out whether the measures of the constructs are culturally invariant, in other words, whether the meaning of the items is the same for respondents in different cultural contexts (Chen et al. 2020; Garcia et al., 2018).

Reply: Measurement validity is not the focus of the current study. However, the issue of measurement invariance has now highlighted in the manuscript, which is written:

“Fourth, measurement invariance of the study variables across different cultural and social contexts is important to ensure external validity of the proposed theoretical structures and findings of the current study [50,51]. Therefore, cross-cultural research is suggested in the future to vindicate the tenability of the influences of family socialization on youths’ psychosocial maturity, as cultural factors profoundly influence of the patterns of family socialization and youth development [4,47].” (Lines 2646-2651, Conclusion)

  1. Yeung, J. W. K. Religion, Family, and Chinese Youth Development. Routledge: London, UK, 2021.
  2. Garcia, F.; Serra, E.; Garcia, O. F.; Martinez, I.; Cruise, E. A third emerging stage for the current digital society? Optimal parenting styles in Spain, the United States, Germany, and Brazil. Int J Environ Res Public Health 2019, 16, 2333.
  3. Chen, F. Z.; Garcia, O. F.; Fuentes, M. C.; Garcia-Ros, R.; Garcia, F. Self-concept in China: Validation of the Chinese version of the Five-Factor Self-Concept (AF5) Questionnaire. Symmetry-Basel 2020, 12, 798.
  4. Garcia, F.; Martinez, I.; Balluerka, N.; Cruise, E.; Garcia, O. F.; Serra, E. Validation of the Five-Factor Self-Concept Questionnaire AF5 in Brazil: Testing factor structure and measurement invariance across language (Brazilian and Spanish), gender, and age. Front Psychol 2018, 9, 2250.

#As limitation it is required include that only one type of parenting (authoritative) is examined.

Reply: the limitation of only examining authoritative parenting in the study is added, which is written:

“Second, this study only examined authoritative parenting in relation to youths’ psychosocial maturity, leaving the influences of other parenting styles and their interaction with family processes on Chinese youth development unknown. In fact, authoritarian parenting was found to bring some benefits to youths in non-Western cultures [48,49], which reveals the importance of investigating and comparing different parenting styles in relation to youth development under the regulation of family processes across different cultural contexts.” (Lines 2636-2642, Conclusion)

  1. Pinquart, M.; Kauser, R. Do the associations of parenting styles with behavior problems and academic achievement vary by culture? Results from a meta-analysis. Cult Divers Ethnic Minor Psychol 2018, 24, 75–100.
  2. Pinquart, M.; Gerke, D. C. Associations of parenting styles with self-esteem in children and adolescents: A meta-analysis. J Child Fam Stud 2019, 28, 2017–2035.

Reviewer 2 Report

- There are too long sentences in the manuscript, and sometimes it is hard to follow them.

- Besides the relevance of information presented in the introduction, it looks like a narrative review, and not an introduction of an original paper. I suggest author to check this point and, if possible, try to summarize it. Furthermore, the relevance of the study should be more clearly presented;

- The aim of the manuscript sounds quite confusing. Please, turn it more objective;

- It is pointed that “if there were two or more children in the same household eligible for the study, e.g. a twin, the older one was selected”. How was identified the “older one” in twin siblings?

- The strategy used to “confidentiality” does not guarantee it;

- Page 6, lines 260-263: this information should be presented in the introduction section;

- What does “intact family” mean? The definition presented on table 1 sounds to be not adequate;

- At the discussion section, author should indicate the reference of the ideas presented;

- It is not clear the conclusion of the study, or even the relevance/implications of its results.

Author Response

Thank you for reviewing the manuscript titled “Family Processes, Parenting Practices, and Psychosocial Maturity of Chinese Youths: A Latent Variable Interaction and Mediation Approach”, and now I have revised the manuscript according to the comments of the reviewers. My responses are below:

Reviewer 2:

# In the text it should be contextualized more detailed  the process of parental socialization. Parenting could help adolescents in their healthy development, but also some parenting practices could not be benefit (Garcia et al., 2020). There are family process within the context of the family such as attachment (Gallarin et al., 2021) or parental expectations (Ridao et al., 2021), so it is important to study parenting combined with other family process.

Reply: Agree, now both family processes and authoritative parenting are contextualized in Chinese context for studying their effects and interaction in contribution to Chinese youths’ development of psychosocial maturity, which is written:

“Family has long been regarded as the most important societal and interpersonal unit affecting individual growth and development in Chinese culture [4,5]. Hence, it is important to scrutinize the relationship between family socialization and Chinese youths’ development of psychosocial maturity.” (Lines 36-40, Introduction)

“Recently, Yeung [2] corroborated the protective effects of youths’ psychosocial maturity on alleviating their internalizing and externalizing symptoms. Thus, as family has been deemed the fundamental and most influential socialization agent in Chinese societies, it is research worthy to scrutinize the relationships of family processes and parenting practices that contribute to Chinese youths’ psychosocial maturity.” (Lines 467-471, The Importance of Psychosocial Maturity to Youth Development)      

“However, extant family research conducted in Chinese societies has predominantly focused on the way parenting behavior affects youth development [22,23], leaving the influence of family processes less examined. As family plays the crucial role of socialization and individual welfare in Chinese culture [4,24], both family processes and parenting practices should be considered equally important when investigating the relationship between family socialization and Chinese youths’ psychosocial maturity. This approach is valid, as family processes highlight the importance of informal socialization by modeling and inculcating family relationships, communication, and interactional dynamics, and parenting practices denote the influence of formal social control by learning and fulfilling parental instructions, requirements, and standards [7,21]. Both correspond to the emphasis of Chinese culture on interpersonal cohesion, attachment, and respect for authority [24,25].” (Lines 481-492, Family Socialization and Psychosocial Maturity)

  1. Yeung, J. W. K., Family socialization and Chinese youth children's development: Does psychosocial maturity matter? Marriage and Family Review 2019, 55, (4), 346-365.
  2. Yeung, J. W. K. Religion, Family, and Chinese Youth Development. Routledge: London, UK, 2021.
  3. Yeung, J. W. K., Investigating the relationships between family socialization and adolescent outcomes in Chinese parent-child dyads. Child Indic Res 2014, 8, 887–905.
  4. Yeung, J. W. K.; Chen, H. F.; Lo, H. H. M.; Choi, A. W. M. Relative effects of parenting practices on child development in the context of family processes. Rev de Psicodidactica 2017, 22, 102–110.
  5. Loke, A. Y.; Mak, Y. W. Family process and peer influences on substance use by adolescents. Int J Environ Res Public Health 2013, 10, (9), 3868-85.
  6. Dou, D. Y.; Shek, D. T. L.; Kwok, K. H. R. Perceived paternal and maternal parenting attributes among Chinese adolescents: A meta-analysis. Int J Environ Res Public Health 2020, 17, 8741.
  7. Liu, Y. R.; Merritt, D. H. Examining the association between parenting and childhood depression among Chinese children and adolescents: A systematic literature review. Child Youth Serv Rev 2018, 88, 316–332.
  8. Fan, H. Y.; Zhang, B. R.; Wang, W. Family functions in relation to behavioral and psychological disorders in Chinese culture. Fam J 2017, 25, 130–136.
  9. Basten, S. The family and social change in Chinese societies. Popul Stud 2015, 69, 127–128.

# The cultural questions should be considered. The impact of parenting on child and adolescent psychosocial could be influenced by the cultural context in which parental socialization take place (e.g., Garcia et al., 2019). More information about parenting studies in China and the particular cultural traits should be included a little bit more detailed.

Reply: As aforementioned, cultural context of Chinese societies has now highlighted and reported in the manuscript, which is written:

“Family has long been regarded as the most important societal and interpersonal unit affecting individual growth and development in Chinese culture [4,5]. Hence, it is important to scrutinize the relationship between family socialization and Chinese youths’ development of psychosocial maturity.” (Lines 36-40, Introduction)

“Recently, Yeung [2] corroborated the protective effects of youths’ psychosocial maturity on alleviating their internalizing and externalizing symptoms. Thus, as family has been deemed the fundamental and most influential socialization agent in Chinese societies, it is research worthy to scrutinize the relationships of family processes and parenting practices that contribute to Chinese youths’ psychosocial maturity.” (Lines 467-471, The Importance of Psychosocial Maturity to Youth Development)      

“However, extant family research conducted in Chinese societies has predominantly focused on the way parenting behavior affects youth development [22,23], leaving the influence of family processes less examined. As family plays the crucial role of socialization and individual welfare in Chinese culture [4,24], both family processes and parenting practices should be considered equally important when investigating the relationship between family socialization and Chinese youths’ psychosocial maturity. This approach is valid, as family processes highlight the importance of informal socialization by modeling and inculcating family relationships, communication, and interactional dynamics, and parenting practices denote the influence of formal social control by learning and fulfilling parental instructions, requirements, and standards [7,21]. Both correspond to the emphasis of Chinese culture on interpersonal cohesion, attachment, and respect for authority [24,25].” (Lines 481-492, Family Socialization and Psychosocial Maturity)

“Second, this study only examined authoritative parenting in relation to youths’ psychosocial maturity, leaving the influences of other parenting styles and their interaction with family processes on Chinese youth development unknown. In fact, authoritarian parenting was found to bring some benefits to youths in non-Western cultures [48,49], which reveals the importance of investigating and comparing different parenting styles in relation to youth development under the regulation of family processes across different cultural contexts.” (Lines 2636-2642, Conclusion)

“Fourth, measurement invariance of the study variables across different cultural and social contexts is important to ensure external validity of the proposed theoretical structures and findings of the current study [50,51]. Therefore, cross-cultural research is suggested in the future to vindicate the tenability of the influences of family socialization on youths’ psychosocial maturity, as cultural factors profoundly influence of the patterns of family socialization and youth development [4,47].”(Lines 2646-2651, Conclusion)

  1. Yeung, J. W. K., Family socialization and Chinese youth children's development: Does psychosocial maturity matter? Marriage and Family Review 2019, 55, (4), 346-365.
  2. Yeung, J. W. K. Religion, Family, and Chinese Youth Development. Routledge: London, UK, 2021.
  3. Yeung, J. W. K., Investigating the relationships between family socialization and adolescent outcomes in Chinese parent-child dyads. Child Indic Res 2014, 8, 887–905.
  4. Yeung, J. W. K.; Chen, H. F.; Lo, H. H. M.; Choi, A. W. M. Relative effects of parenting practices on child development in the context of family processes. Rev de Psicodidactica 2017, 22, 102–110.
  5. Loke, A. Y.; Mak, Y. W. Family process and peer influences on substance use by adolescents. Int J Environ Res Public Health 2013, 10, (9), 3868-85.
  6. Dou, D. Y.; Shek, D. T. L.; Kwok, K. H. R. Perceived paternal and maternal parenting attributes among Chinese adolescents: A meta-analysis. Int J Environ Res Public Health 2020, 17, 8741.
  7. Liu, Y. R.; Merritt, D. H. Examining the association between parenting and childhood depression among Chinese children and adolescents: A systematic literature review. Child Youth Serv Rev 2018, 88, 316–332.
  8. Fan, H. Y.; Zhang, B. R.; Wang, W. Family functions in relation to behavioral and psychological disorders in Chinese culture. Fam J 2017, 25, 130–136.
  9. Basten, S. The family and social change in Chinese societies. Popul Stud 2015, 69, 127–128.
  10. Garcia, F.; Serra, E.; Garcia, O. F.; Martinez, I.; Cruise, E. A third emerging stage for the current digital society? Optimal parenting styles in Spain, the United States, Germany, and Brazil. Int J Environ Res Public Health 2019, 16, 2333.
  11. Pinquart, M.; Kauser, R. Do the associations of parenting styles with behavior problems and academic achievement vary by culture? Results from a meta-analysis. Cult Divers Ethnic Minor Psychol 2018, 24, 75–100.
  12. Pinquart, M.; Gerke, D. C. Associations of parenting styles with self-esteem in children and adolescents: A meta-analysis. J Child Fam Stud 2019, 28, 2017–2035.
  13. Chen, F. Z.; Garcia, O. F.; Fuentes, M. C.; Garcia-Ros, R.; Garcia, F. Self-concept in China: Validation of the Chinese version of the Five-Factor Self-Concept (AF5) Questionnaire. Symmetry-Basel 2020, 12, 798.
  14. Garcia, F.; Martinez, I.; Balluerka, N.; Cruise, E.; Garcia, O. F.; Serra, E. Validation of the Five-Factor Self-Concept Questionnaire AF5 in Brazil: Testing factor structure and measurement invariance across language (Brazilian and Spanish), gender, and age. Front Psychol 2018, 9, 2250.

Reviewer 3 Report

This study analyzes parenting and family functioning in China. Along with the impact of parental socialization, the family climate is also examined. Parenting only includes a measure about authoritative parenting, but not other types of parenting. Family climate and authoritative parenting are positively related to adolescent psychosocial competence.

In the text it should be contextualized more detailed  the process of parental socialization. Parenting could help adolescents in their healthy development, but also some parenting practices could not be benefit (Garcia et al., 2020). There are family process within the context of the family such as attachment (Gallarin et al., 2021) or parental expectations (Ridao et al., 2021), so it is important to study parenting combined with other family process.

The cultural questions should be considered. The impact of parenting on child and adolescent psychosocial could be influenced by the cultural context in which parental socialization take place (e.g., Garcia et al., 2019). More information about parenting studies in China and the particular cultural traits should be included a little bit more detailed.

References

Gallarin, M., Torres-Gomez, B., & Alonso-Arbiol, I. (2021). Aggressiveness in adopted and non-adopted teens: The role of parenting, attachment security, and gender. International Journal of Environmental Research and Public Health, 18(2034), 1-16. doi:10.3390/ijerph18042034

Garcia, F., Serra, E., Garcia, O. F., Martinez, I., & Cruise, E. (2019). A third emerging stage for the current digital society? Optimal parenting styles in Spain, the United States, Germany, and Brazil. International Journal of Environmental Research and Public Health, 16(2333), 1-20. doi:10.3390/ijerph16132333

Garcia, O. F., Fuentes, M. C., Gracia, E., Serra, E., & Garcia, F. (2020). Parenting warmth and strictness across three generations: Parenting styles and psychosocial adjustment. International Journal of Environmental Research and Public Health, 17, 1-18. doi:10.3390/ijerph17207487

Ridao, P., López-Verdugo, I., & Reina-Flores, C. (2021). Parental beliefs about childhood and adolescence from a longitudinal perspective. International Journal of Environmental Research and Public Health, 18(1760), 1-17. doi:10.3390/ijerph18041760

Author Response

Thank you for reviewing the manuscript titled “Family Processes, Parenting Practices, and Psychosocial Maturity of Chinese Youths: A Latent Variable Interaction and Mediation Approach”, and now I have revised the manuscript according to the comments of the reviewers. My responses are below:

Reviewer 3:

# There are too long sentences in the manuscript, and sometimes it is hard to follow them.

Reply: The current study attempts to examine the effects of family processes and parenting practices and their latent interaction on Chinese youths’ psychosocial maturity with a latent variable modeling approach. Hence, due to the relatively complicated study relationships between family processes, parenting practices, and Chinese youths’ psychosocial maturity needed to be explicated, a little bit dedicated sentence structures are unavoidable. However, I have now reviewed and revised the whole manuscript again for trying to make the presentations more concise and accurate. Further, a native English expert in Canada has helped to copy-edit the manuscript for assuring the grammatical and sentence structures correct.       

# Besides the relevance of information presented in the introduction, it looks like a narrative review, and not an introduction of an original paper. I suggest author to check this point and, if possible, try to summarize it. Furthermore, the relevance of the study should be more clearly presented;

Reply: Now the Introduction has been completely revised and presented in a more direct and concise way to present the importance of studying the effects of family socialization by family processes and parenting practices on Chinese youths’ psychosocial maturity with a latent variable modeling approach.

# The aim of the manuscript sounds quite confusing. Please, turn it more objective;

Reply: The study aim is now presented more directly and precisely in Introduction, and a new section “2.4. The Present Study” is added to summarize the study relationships between positive family processes, authoritative parenting, and Chinese youths’ psychosocial maturity, in which THREE hypotheses are set up to help readers clearly grasp the structural relationships between positive family processes, authoritative parenting, and Chinese youths’ psychosocial maturity. They are listed below

“In sum, this study intends to investigate the way family processes and parenting practices and their interaction commonly contribute to the psychosocial maturity of Chinese youths. Parenting practices are believed to mediate the relationship between family processes and Chinese youths’ psychosocial maturity, and family processes would interact with parenting practices to predict Chinese youths’ psychosocial maturity by moderating the mediating effect of parenting practices on Chinese youths’ psychosocial maturity.” (Lines 222-228, Introduction)  

“2.4. The Present Study

The current study takes a latent variable approach to investigate the effects of positive family processes and authoritative parenting and their interaction on Chinese youths’ psychosocial maturity. Authoritative parenting is expected to mediate the effect of positive family processes on Chinese youths’ psychosocial maturity, and positive family processes would moderate the mediating effect of authoritative parenting on Chinese youths’ psychosocial maturity. Accordingly, the hypotheses are set below.

H1. Both positive family processes and authoritative parenting and their latent interaction would positively predict Chinese youths’ psychosocial maturity.

H2. Authoritative parenting would mediate the relationship between positive family processes and Chinese youths’ psychosocial maturity.

H3. Positive family processes would moderate the mediating effect of authoritative parenting on Chinese youths’ psychosocial maturity, in which the strongest mediating effect is expected in high positive family processes, and the least strong mediating effect is expected in low positive family processes, and the moderate mediating effect is expected in medium positive family processes.

The sociodemographic covariates of family composition, family welfare dependency, youth gender, and age were adjusted in the study relationships because prior research demonstrated their effects on youth development. Specifically, family composition refers to whether the youth participant lives with both biological mother and father or not (intact family vs otherwise). Family welfare dependency indicates whether the participating family receives any financial subsidies from the government. Youths living with both biological mother and father, referring to intact family structure, and without welfare dependency would exhibit better development [4,20,31]. Being female and older would exhibit more maturity and considerateness [4,32]. In the modeling procedures, family composition, family welfare dependency, youth gender and age were adjusted as control variables to preclude their confounding effects.” (Lines 1098-1129, The Present Study)

  1. Yeung, J. W. K., Religion, Family, and Chinese Youth Development. Routledge: London, 2021.
  2. Loke, A. Y.; Mak, Y. W., Family process and peer influences on substance use by adolescents. Int J Environ Res Public Health 2013, 10, (9), 3868-85.
  3. Regnerus, M. D.; Elder, G. H., Religion and vulnerability among low-risk adolescents. Social Science Research 2003, 32, (4), 633-658.
  4. Wills, T. A.; Gibbons, F. X.; Gerrard, M.; Murry, V. M.; Brody, G. H., Family communication and religiosity related to substance use and sexual behavior in early adolescence: A test for pathways through self-control and prototype perceptions. Psychology of Addictive Behaviors 2003, 17, (4), 312-323.

# It is pointed that “if there were two or more children in the same household eligible for the study, e.g. a twin, the older one was selected”. How was identified the “older one” in twin siblings?

Reply: The presentation is now rephrased to “However, if there were two or more children in the same household eligible for the study, a twin for example, the first born one was selected.” (Lines 1329-1330)

#The strategy used to “confidentiality” does not guarantee it;

Reply: Agree, the author could not guarantee confidentiality, but needs to devise procedures to seek for privacy. Hence, now the sentence is rewritten to “To ensure personal privacy, questionnaires were contained in independent envelops separately for the parent and child participants, and they returned their completed questionnaires in the same envelops.” (Lines 1330-1333)

#Page 6, lines 260-263: this information should be presented in the introduction section;

Reply: Lines 260-263 serve to present the need of controlling confounding the effects of family composition, family welfare dependency, youth gender and age on Chinese youths’ psychosocial maturity. Now, this presentation has moved to the newly added section “2.4. The Present Study” to let readers more closely follow the importance of controlling them as covariates in the study relationships, which is written

“The sociodemographic covariates of family composition, family welfare dependency, youth gender, and age were adjusted in the study relationships because prior research demonstrated their effects on youth development. Specifically, family composition refers to whether the youth participant lives with both biological mother and father or not (intact family vs otherwise). Family welfare dependency indicates whether the participating family receives any financial subsidies from the government. Youths living with both biological mother and father, referring to intact family structure, and without welfare dependency would exhibit better development [4,20,31]. Being female and older would exhibit more maturity and considerateness [4,32]. In the modeling procedures, family composition, family welfare dependency, youth gender and age were adjusted as control variables to preclude their confounding effects.” (Lines 1118-1128, Methods)

  1. Yeung, J. W. K., Religion, Family, and Chinese Youth Development. Routledge: London, 2021.
  2. Loke, A. Y.; Mak, Y. W., Family process and peer influences on substance use by adolescents. Int J Environ Res Public Health 2013, 10, (9), 3868-85.
  3. Regnerus, M. D.; Elder, G. H., Religion and vulnerability among low-risk adolescents. Social Science Research 2003, 32, (4), 633-658.
  4. Wills, T. A.; Gibbons, F. X.; Gerrard, M.; Murry, V. M.; Brody, G. H., Family communication and religiosity related to substance use and sexual behavior in early adolescence: A test for pathways through self-control and prototype perceptions. Psychology of Addictive Behaviors 2003, 17, (4), 312-323.

#What does “intact family” mean? The definition presented on table 1 sounds to be not adequate;

Reply: Intact family is now more clearly defined as “family composition refers to whether the youth participant lives with both biological mother and father or not (intact family vs otherwise)” (Lines 1120-1122) and “Youths living with both biological mother and father, referring to intact family structure”. (Lines 1123-1124)

#At the discussion section, author should indicate the reference of the ideas presented;

Reply: The Discussion is now revised and incorporate more relevant references regarding to the ideas presented.

#It is not clear the conclusion of the study, or even the relevance/implications of its results.

Reply: A new section of “Conclusion” is added to summarize the main findings and contributions of the study, and limitations of the study that originally presented in the section of “Discussion” are now moved to “Conclusion” for the coherence of the flows of the manuscript.

Round 2

Reviewer 1 Report

Authors have considered all the suggestions made in the previous revision, including in the manuscript the opportune modifications.

a) It has been incorporated the suggested terminology (authoritative parenting)

b) Both in the Introduction and in the Conclusion of the paper, has been highlighted the importance to test the influence of parenting on adolescent adjustement and competence across different cultural contexts. Furthermore, it has been included the suggested citations/references. 

c) In the Discussion section has been incorporated the idea that adolescence is a critical developmental time related to some vulnerability and difficulties, in part due to not all adolescents have enough psychosocial maturity. 

c) Sample description, sampling procedures and selection criteria has been successfully incorporated

d)  The table of correlations between the study variables has been incorporated, as well as the appropriate comments on it.

e) Finally, among the limitations of the study, authors stand out that only one type of parenting (authoritative) has been examined, highlighting the importance to analyze the effects of the different parenting styles on the youth development across different cultural contexts.

In consequence, it’s suggested that this version of the manuscript could be adequate for publication. 

Author Response

Dear Dr Oscar Fernando García,

Thank you for giving feedback for the submitted manuscript titled “Family Processes, Parenting Practices, and Psychosocial Maturity of Chinese Youths: A Latent Variable Interaction and Mediation Approach”, and now I have revised the manuscript according to the comments of the three reviewers. My responses are below:

For reviewer 1:

a) It has been incorporated the suggested terminology (authoritative parenting)

Reply: Yes and agree

b)Both in the Introduction and in the Conclusion of the paper, has been highlighted the importance to test the influence of parenting on adolescent adjustement and competence across different cultural contexts. Furthermore, it has been included the suggested citations/references.

Reply: Yes and thank you for the suggestion of the cultural factor in sway of youth development.

c)In the Discussion section has been incorporated the idea that adolescence is a critical developmental time related to some vulnerability and difficulties, in part due to not all adolescents have enough psychosocial maturity. 

Reply: Yes, and again thank you for reminding the need to highlight possible challenges of adolescence to some youths.

d)Sample description, sampling procedures and selection criteria has been successfully incorporated

Reply: Yes, now more details of the sampling procedures and selection criteria are reported.

e)The table of correlations between the study variables has been incorporated, as well as the appropriate comments on it.

Reply: Agree

f)Finally, among the limitations of the study, authors stand out that only one type of parenting (authoritative) has been examined, highlighting the importance to analyze the effects of the different parenting styles on the youth development across different cultural contexts.

Reply: Yes and thank you for the reminder of the limitation by only employing “authoritative parenting” as one type of parenting in the study.

In consequence, it’s suggested that this version of the manuscript could be adequate for publication. 

Reply: Thank you for your suggestion of accepting the manuscript for publication

As the comments of reviewer 2 and 3 and my responses are related to the manuscript, hence I enclosed as below:

For reviewer 2:

# Authors explained the meaning of “intact family”, and I really think they should try another expression, given its use may sound prejudiced (it is known there are different families structures, and designate one as “intact”, means that the others are “not ideal”). I suggest, for example, the use of other expressions, such as “two biological parents”, instead of “intact family”.

Reply: Now the term of “intact family” to describe Chinese youths living with both biological mother and father has changed to “two biological parents family”, and I agree the reviewer’s suggestion to try to use a more neutral and acceptable term to avoid any unnecessary discrimination.

# When authors present the descriptive information/distribution of variables, sometimes they present it as percentage, sometimes they present in absolute frequency. I suggest them to standardize it. Furthermore, results description sounds, sometimes, quite repetitive (e.g. description of table 2 – authors should try to present the information without so many repetitions).

Reply: Now the descriptive information/distribution of variables in the Results have been presented in percentages, and also the description of correlations of table 2 has been trimmed and made more concise presentation, for which they are now written as below:

“Table 1 presents the sociodemographic characteristics of the parent-youth dyads: 88.4% came from two biological parent family, and 11.6% were from other family structure; and 43.9% of the participating families were welfare-dependent, and 56.1% were non-dependent. The gender of the main caregiver parents was 80.1% mothers and 19.9% fathers. Also, 57.4% of the youth participants were female and 42.6% were male. Their average age was 16.30, meaning generally in middle adolescence.”

“Table 2 shows the correlation coefficients of the main study variables, in which positive family processes were substantially correlated with authoritative parenting, r=.744, p<.001. Moreover, positive family processes were significantly correlated with youth self-concept, self-control, and perspective taking, r= .323, .263, and .280, p< .001. Authoritative parenting was significantly correlated with youth self-concept, self-control, and perspective taking, r= .242, .283, and .315, p< .001. Besides, youth self-concept, self-control, and perspective taking were significantly and concretely correlated with each other, ranging from r=.363 to .481, p< .001.”

# I authors should check the manuscript for possible typos

Reply: The whole manuscript has now completely and carefully proofread to avoid any typos.                                  

For reviewer 3:

# However, in the discussion it should be added more details including other family variables that could be examined along with the impact of parenting such as parental beliefs (Ridao, Lopez-Verdugo, & Reina-Flores, 2021) or attachment (Gallarin, Torres-Gomez, & Alonso-Arbiol, 2021). This point is important to suggest future studies similar to the line followed in present study.

Reply: Agree, and I have now highlighted the importance of incorporating other family variables in work with parenting practices and family processes to affect youth development for future research in the Discussion, which is written as

“Besides, as aforementioned family socialization is a multifaceted concept, in which parental beliefs and parent-youth attachment as well as emotional expressions of family members are all influential of youth development [45-47]. However, we currently know little about how these cognitive and relational dimensions of family socialization in work with different parenting styles and family processes to shape youth development [4, 6, 48]. Thus, to take an integrative approach to investigate the effects of different family socialization facets and their interactive effects on youth development concomitantly is important to enhance our understanding the relationship between family socialization and youth development.”

4. Yeung, J. W. K., Religion, Family, and Chinese Youth Development. Routledge: London, 2021.

6.Lorenzo-Blanco, E. I.; Bares, C. B.; Delva, J., Parenting, family processes, relationships, and parental support in multiracial and multiethnic families: An exploratory study of youth perceptions. Family Relations 2013, 62, (1), 125-139.

45.Ridao, P.; Lopez-Verdugo, I.; Reina-Flores, C., Parental Beliefs about Childhood and Adolescence from a Longitudinal Perspective. International Journal of Environmental Research and Public Health 2021, 18, (4), 1760.

46.Garcia, O. F.; Fuentes, M. C.; Gracia, E.; Serra, E.; Garcia, F., Parenting warmth and strictness across three generations: Parenting styles and psychosocial adjustment. International Journal of Environmental Research and Public Health 2020, 17, (20), 7487.

47.Garcia, F.; Serra, E.; Garcia, O. F.; Martinez, I.; Cruise, E., A Third Emerging Stage for the Current Digital Society? Optimal Parenting Styles in Spain, the United States, Germany, and Brazil. International Journal of Environmental Research and Public Health 2019, 16, (13), 2333.

48.Garcia, F.; Serra, E.; Garcia, O. F.; Martinez, I.; Cruise, E., A Third Emerging Stage for the Current Digital Society? Optimal Parenting Styles in Spain, the United States, Germany, and Brazil. International Journal of Environmental Research and Public Health 2019, 16, (13), 2333.

# Additionally, a relevant finding of present study is the constantly correlation of the parenting style captured in this study (authoritative) with the other variables of the family and the adolescent psychosocial development. It should be discussed considering the relevance of parenting styles as the emotional family climate, which represents relational qualities between parents and their children. (Garcia, Fuentes, Gracia, Serra, & Garcia, 2020; Martinez, Murgui, Garcia, & Garcia, 2021).

Reply: Now the importance of emotional family climate has been mentioned in the Discussion, and for coherence we mentioned this point in alignment with the point of incorporating other family variables highlighted above, which can hence be viewed in the reply above.

# Authors should use the label *authoritative* along the text. In the Figure 2, authors should modify *effective parenting practices* by *authoritative*.

Reply: Now, “authoritative parenting” has been used to replace “effective parenting practices” in Figure 1 and 2.

Finally, I am greatly thankful for the careful and thoughtful comments and suggestions of the reviewers in the reviewing process.

Best

The author

Reviewer 2 Report

- Authors explained the meaning of “intact family”, and I really think they should try another expression, given its use may sound prejudiced (it is known there are different families structures, and designate one as “intact”, means that the others are “not ideal”). I suggest, for example, the use of other expressions, such as “two biological parents”, instead of “intact family”.

- When authors present the descriptive information/distribution of variables, sometimes they present it as percentage, sometimes they present in absolute frequency. I suggest them to standardize it. Furthermore, results description sounds, sometimes, quite repetitive (e.g. description of table 2 – authors should try to present the information without so many repetitions).

- I authors should check the manuscript for possible typos.

Author Response

Dear Dr Oscar Fernando García,

Thank you for giving feedback for the submitted manuscript titled “Family Processes, Parenting Practices, and Psychosocial Maturity of Chinese Youths: A Latent Variable Interaction and Mediation Approach”, and now I have revised the manuscript according to the comments of the three reviewers. My responses are below:

For reviewer 2:

# Authors explained the meaning of “intact family”, and I really think they should try another expression, given its use may sound prejudiced (it is known there are different families structures, and designate one as “intact”, means that the others are “not ideal”). I suggest, for example, the use of other expressions, such as “two biological parents”, instead of “intact family”.

Reply: Now the term of “intact family” to describe Chinese youths living with both biological mother and father has changed to “two biological parents family”, and I agree the reviewer’s suggestion to try to use a more neutral and acceptable term to avoid any unnecessary discrimination.

# When authors present the descriptive information/distribution of variables, sometimes they present it as percentage, sometimes they present in absolute frequency. I suggest them to standardize it. Furthermore, results description sounds, sometimes, quite repetitive (e.g. description of table 2 – authors should try to present the information without so many repetitions).

Reply: Now the descriptive information/distribution of variables in the Results have been presented in percentages, and also the description of correlations of table 2 has been trimmed and made more concise presentation, for which they are now written as below:

“Table 1 presents the sociodemographic characteristics of the parent-youth dyads: 88.4% came from two biological parent family, and 11.6% were from other family structure; and 43.9% of the participating families were welfare-dependent, and 56.1% were non-dependent. The gender of the main caregiver parents was 80.1% mothers and 19.9% fathers. Also, 57.4% of the youth participants were female and 42.6% were male. Their average age was 16.30, meaning generally in middle adolescence.”

“Table 2 shows the correlation coefficients of the main study variables, in which positive family processes were substantially correlated with authoritative parenting, r=.744, p<.001. Moreover, positive family processes were significantly correlated with youth self-concept, self-control, and perspective taking, r= .323, .263, and .280, p< .001. Authoritative parenting was significantly correlated with youth self-concept, self-control, and perspective taking, r= .242, .283, and .315, p< .001. Besides, youth self-concept, self-control, and perspective taking were significantly and concretely correlated with each other, ranging from r=.363 to .481, p< .001.”

# I authors should check the manuscript for possible typos

Reply: The whole manuscript has now completely and carefully proofread to avoid any typos.                                  

As the comments of reviewer 1 and 3 and my responses are related to the manuscript, hence I enclosed as below:

For reviewer 1:

a) It has been incorporated the suggested terminology (authoritative parenting)

Reply: Yes and agree

b)Both in the Introduction and in the Conclusion of the paper, has been highlighted the importance to test the influence of parenting on adolescent adjustement and competence across different cultural contexts. Furthermore, it has been included the suggested citations/references.

Reply: Yes and thank you for the suggestion of the cultural factor in sway of youth development.

c)In the Discussion section has been incorporated the idea that adolescence is a critical developmental time related to some vulnerability and difficulties, in part due to not all adolescents have enough psychosocial maturity. 

Reply: Yes, and again thank you for reminding the need to highlight possible challenges of adolescence to some youths.

d)Sample description, sampling procedures and selection criteria has been successfully incorporated

Reply: Yes, now more details of the sampling procedures and selection criteria are reported.

e)The table of correlations between the study variables has been incorporated, as well as the appropriate comments on it.

Reply: Agree

f)Finally, among the limitations of the study, authors stand out that only one type of parenting (authoritative) has been examined, highlighting the importance to analyze the effects of the different parenting styles on the youth development across different cultural contexts.

Reply: Yes and thank you for the reminder of the limitation by only employing “authoritative parenting” as one type of parenting in the study.

In consequence, it’s suggested that this version of the manuscript could be adequate for publication. 

Reply: Thank you for your suggestion of accepting the manuscript for publication

For reviewer 3:

# However, in the discussion it should be added more details including other family variables that could be examined along with the impact of parenting such as parental beliefs (Ridao, Lopez-Verdugo, & Reina-Flores, 2021) or attachment (Gallarin, Torres-Gomez, & Alonso-Arbiol, 2021). This point is important to suggest future studies similar to the line followed in present study.

Reply: Agree, and I have now highlighted the importance of incorporating other family variables in work with parenting practices and family processes to affect youth development for future research in the Discussion, which is written as

“Besides, as aforementioned family socialization is a multifaceted concept, in which parental beliefs and parent-youth attachment as well as emotional expressions of family members are all influential of youth development [45-47]. However, we currently know little about how these cognitive and relational dimensions of family socialization in work with different parenting styles and family processes to shape youth development [4, 6, 48]. Thus, to take an integrative approach to investigate the effects of different family socialization facets and their interactive effects on youth development concomitantly is important to enhance our understanding the relationship between family socialization and youth development.”

4. Yeung, J. W. K., Religion, Family, and Chinese Youth Development. Routledge: London, 2021.

6.Lorenzo-Blanco, E. I.; Bares, C. B.; Delva, J., Parenting, family processes, relationships, and parental support in multiracial and multiethnic families: An exploratory study of youth perceptions. Family Relations 2013, 62, (1), 125-139.

45.Ridao, P.; Lopez-Verdugo, I.; Reina-Flores, C., Parental Beliefs about Childhood and Adolescence from a Longitudinal Perspective. International Journal of Environmental Research and Public Health 2021, 18, (4), 1760.

46.Garcia, O. F.; Fuentes, M. C.; Gracia, E.; Serra, E.; Garcia, F., Parenting warmth and strictness across three generations: Parenting styles and psychosocial adjustment. International Journal of Environmental Research and Public Health 2020, 17, (20), 7487.

47.Garcia, F.; Serra, E.; Garcia, O. F.; Martinez, I.; Cruise, E., A Third Emerging Stage for the Current Digital Society? Optimal Parenting Styles in Spain, the United States, Germany, and Brazil. International Journal of Environmental Research and Public Health 2019, 16, (13), 2333.

48.Garcia, F.; Serra, E.; Garcia, O. F.; Martinez, I.; Cruise, E., A Third Emerging Stage for the Current Digital Society? Optimal Parenting Styles in Spain, the United States, Germany, and Brazil. International Journal of Environmental Research and Public Health 2019, 16, (13), 2333.

# Additionally, a relevant finding of present study is the constantly correlation of the parenting style captured in this study (authoritative) with the other variables of the family and the adolescent psychosocial development. It should be discussed considering the relevance of parenting styles as the emotional family climate, which represents relational qualities between parents and their children. (Garcia, Fuentes, Gracia, Serra, & Garcia, 2020; Martinez, Murgui, Garcia, & Garcia, 2021).

Reply: Now the importance of emotional family climate has been mentioned in the Discussion, and for coherence we mentioned this point in alignment with the point of incorporating other family variables highlighted above, which can hence be viewed in the reply above.

# Authors should use the label *authoritative* along the text. In the Figure 2, authors should modify *effective parenting practices* by *authoritative*.

Reply: Now, “authoritative parenting” has been used to replace “effective parenting practices” in Figure 1 and 2.

Finally, I am greatly thankful for the careful and thoughtful comments and suggestions of the reviewers in the reviewing process.

Best

The author

Reviewer 3 Report

Authors have made some modifications that have improved the quality of the manuscript. In present manuscript is examined parenting along with family processes to identify their impact on adolescent psychosocial maturity.

However, in the discussion it should be added more details including other family variables that could be examined along with the impact of parenting such as parental beliefs (Ridao, Lopez-Verdugo, & Reina-Flores, 2021) or attachment (Gallarin, Torres-Gomez, & Alonso-Arbiol, 2021). This point is important to suggest future studies similar to the line followed in present study.

Additionally, a relevant finding of present study is the constantly correlation of the parenting style captured in this study (authoritative) with the other variables of the family and the adolescent psychosocial development. It should be discussed considering the relevance of parenting styles as the emotional family climate, which represents relational qualities between parents and their children. (Garcia, Fuentes, Gracia, Serra, & Garcia, 2020; Martinez, Murgui, Garcia, & Garcia, 2021).

Authors should use the label *authoritative* along the text. In the Figure 2, authors should modify *effective parenting practices* by *authoritative*.

References

Gallarin, M., Torres-Gomez, B., & Alonso-Arbiol, I. (2021). Aggressiveness in adopted and non-adopted teens: The role of parenting, attachment security, and gender. International Journal of Environmental Research and Public Health, 18(2034), 1-16. doi:10.3390/ijerph18042034

Garcia, O. F., Fuentes, M. C., Gracia, E., Serra, E., & Garcia, F. (2020). Parenting warmth and strictness across three generations: Parenting styles and psychosocial adjustment. International Journal of Environmental Research and Public Health, 17, 1-18. doi:10.3390/ijerph17207487

Martinez, I., Murgui, S., Garcia, O. F., & Garcia, F. (2021). Parenting and adolescent adjustment: The mediational role of family self-esteem. Journal of Child and Family Studies. doi:10.1007/s10826-021-01937-z

Ridao, P., López-Verdugo, I., & Reina-Flores, C. (2021). Parental beliefs about childhood and adolescence from a longitudinal perspective. International Journal of Environmental Research and Public Health, 18(1760), 1-17. doi:10.3390/ijerph18041760

Author Response

Dear Dr Oscar Fernando García,

Thank you for giving feedback for the submitted manuscript titled “Family Processes, Parenting Practices, and Psychosocial Maturity of Chinese Youths: A Latent Variable Interaction and Mediation Approach”, and now I have revised the manuscript according to the comments of the three reviewers. My responses are below:

For Reviewer 3

# However, in the discussion it should be added more details including other family variables that could be examined along with the impact of parenting such as parental beliefs (Ridao, Lopez-Verdugo, & Reina-Flores, 2021) or attachment (Gallarin, Torres-Gomez, & Alonso-Arbiol, 2021). This point is important to suggest future studies similar to the line followed in present study.

Reply: Agree, and I have now highlighted the importance of incorporating other family variables in work with parenting practices and family processes to affect youth development for future research in the Discussion, which is written as

“Besides, as aforementioned family socialization is a multifaceted concept, in which parental beliefs and parent-youth attachment as well as emotional expressions of family members are all influential of youth development [45-47]. However, we currently know little about how these cognitive and relational dimensions of family socialization in work with different parenting styles and family processes to shape youth development [4, 6, 48]. Thus, to take an integrative approach to investigate the effects of different family socialization facets and their interactive effects on youth development concomitantly is important to enhance our understanding the relationship between family socialization and youth development.”

4. Yeung, J. W. K., Religion, Family, and Chinese Youth Development. Routledge: London, 2021.

6.Lorenzo-Blanco, E. I.; Bares, C. B.; Delva, J., Parenting, family processes, relationships, and parental support in multiracial and multiethnic families: An exploratory study of youth perceptions. Family Relations 2013, 62, (1), 125-139.

45.Ridao, P.; Lopez-Verdugo, I.; Reina-Flores, C., Parental Beliefs about Childhood and Adolescence from a Longitudinal Perspective. International Journal of Environmental Research and Public Health 2021, 18, (4), 1760.

46.Garcia, O. F.; Fuentes, M. C.; Gracia, E.; Serra, E.; Garcia, F., Parenting warmth and strictness across three generations: Parenting styles and psychosocial adjustment. International Journal of Environmental Research and Public Health 2020, 17, (20), 7487.

47.Garcia, F.; Serra, E.; Garcia, O. F.; Martinez, I.; Cruise, E., A Third Emerging Stage for the Current Digital Society? Optimal Parenting Styles in Spain, the United States, Germany, and Brazil. International Journal of Environmental Research and Public Health 2019, 16, (13), 2333.

48.Garcia, F.; Serra, E.; Garcia, O. F.; Martinez, I.; Cruise, E., A Third Emerging Stage for the Current Digital Society? Optimal Parenting Styles in Spain, the United States, Germany, and Brazil. International Journal of Environmental Research and Public Health 2019, 16, (13), 2333.

# Additionally, a relevant finding of present study is the constantly correlation of the parenting style captured in this study (authoritative) with the other variables of the family and the adolescent psychosocial development. It should be discussed considering the relevance of parenting styles as the emotional family climate, which represents relational qualities between parents and their children. (Garcia, Fuentes, Gracia, Serra, & Garcia, 2020; Martinez, Murgui, Garcia, & Garcia, 2021).

Reply: Now the importance of emotional family climate has been mentioned in the Discussion, and for coherence we mentioned this point in alignment with the point of incorporating other family variables highlighted above, which can hence be viewed in the reply above.

# Authors should use the label *authoritative* along the text. In the Figure 2, authors should modify *effective parenting practices* by *authoritative*.

Reply: Now, “authoritative parenting” has been used to replace “effective parenting practices” in Figure 1 and 2.

Finally, I am greatly thankful for the careful and thoughtful comments and suggestions of the reviewers in the reviewing process.

As the comments of reviewer 1 and 2 and my responses are related to the manuscript, hence I enclosed as below:

For reviewer 1:

a) It has been incorporated the suggested terminology (authoritative parenting)

Reply: Yes and agree

b)Both in the Introduction and in the Conclusion of the paper, has been highlighted the importance to test the influence of parenting on adolescent adjustement and competence across different cultural contexts. Furthermore, it has been included the suggested citations/references.

Reply: Yes and thank you for the suggestion of the cultural factor in sway of youth development.

c)In the Discussion section has been incorporated the idea that adolescence is a critical developmental time related to some vulnerability and difficulties, in part due to not all adolescents have enough psychosocial maturity. 

Reply: Yes, and again thank you for reminding the need to highlight possible challenges of adolescence to some youths.

d)Sample description, sampling procedures and selection criteria has been successfully incorporated

Reply: Yes, now more details of the sampling procedures and selection criteria are reported.

e)The table of correlations between the study variables has been incorporated, as well as the appropriate comments on it.

Reply: Agree

f)Finally, among the limitations of the study, authors stand out that only one type of parenting (authoritative) has been examined, highlighting the importance to analyze the effects of the different parenting styles on the youth development across different cultural contexts.

Reply: Yes and thank you for the reminder of the limitation by only employing “authoritative parenting” as one type of parenting in the study.

In consequence, it’s suggested that this version of the manuscript could be adequate for publication. 

Reply: Thank you for your suggestion of accepting the manuscript for publication

For Reviewer 2

# Authors explained the meaning of “intact family”, and I really think they should try another expression, given its use may sound prejudiced (it is known there are different families structures, and designate one as “intact”, means that the others are “not ideal”). I suggest, for example, the use of other expressions, such as “two biological parents”, instead of “intact family”.

Reply: Now the term of “intact family” to describe Chinese youths living with both biological mother and father has changed to “two biological parents family”, and I agree the reviewer’s suggestion to try to use a more neutral and acceptable term to avoid any unnecessary discrimination.

# When authors present the descriptive information/distribution of variables, sometimes they present it as percentage, sometimes they present in absolute frequency. I suggest them to standardize it. Furthermore, results description sounds, sometimes, quite repetitive (e.g. description of table 2 – authors should try to present the information without so many repetitions).

Reply: Now the descriptive information/distribution of variables in the Results have been presented in percentages, and also the description of correlations of table 2 has been trimmed and made more concise presentation, for which they are now written as below:

“Table 1 presents the sociodemographic characteristics of the parent-youth dyads: 88.4% came from two biological parent family, and 11.6% were from other family structure; and 43.9% of the participating families were welfare-dependent, and 56.1% were non-dependent. The gender of the main caregiver parents was 80.1% mothers and 19.9% fathers. Also, 57.4% of the youth participants were female and 42.6% were male. Their average age was 16.30, meaning generally in middle adolescence.”

“Table 2 shows the correlation coefficients of the main study variables, in which positive family processes were substantially correlated with authoritative parenting, r=.744, p<.001. Moreover, positive family processes were significantly correlated with youth self-concept, self-control, and perspective taking, r= .323, .263, and .280, p< .001. Authoritative parenting was significantly correlated with youth self-concept, self-control, and perspective taking, r= .242, .283, and .315, p< .001. Besides, youth self-concept, self-control, and perspective taking were significantly and concretely correlated with each other, ranging from r=.363 to .481, p< .001.”

# I authors should check the manuscript for possible typos

Reply: The whole manuscript has now completely and carefully proofread to avoid any typos.                                  

Best

The author

Round 3

Reviewer 3 Report

I have checked the reviewed version and I only have seen only very minor

Points.

Authors have made important modifications that considerably improve the

quality of the manuscript. Only some minor points should be addressed.

Statistics (p, r, etc.) should be in italic along the text.

In table 2, in the main diagonal, it seems not completed, it should

appear “--” or 1.

Additionally, in the discussion section or in conclusion section, in

order to connect the present study with future studies, it should be

highlighted a little bit more the relevance of mediational variables in

order to explain the impact of parenting on child and adolescent

psychosocial maturity (e.g., Martinez et al. 2021).

References:

Martinez, I., Murgui, S., Garcia, O. F., & Garcia, F. (2021). Parenting

and adolescent adjustment: The mediational role of family self-esteem.

Journal of Child and Family Studies. doi:10.1007/s10826-021-01937-z

Best regards,

Author Response

For reviewer 3:

#  Statistics (p, r, etc.) should be in italic along the text.  In table 2, in the main diagonal, it seems not completed, it should

appear “--” or 1.

Reply: Now all the statistics and their symbols are presented in italics, and a symbol of -- has put in the correction between the same variable in Table 2.

#  Additionally, in the discussion section or in conclusion section, in order to connect the present study with future studies, it should be highlighted a little bit more the relevance of mediational variables in order to explain the impact of parenting on child and adolescent psychosocial maturity (e.g., Martinez et al. 2021).

Reply: Now the suggestion of the mediation of parenting and its related variables has cited in the Discussion, which is written

"Authoritative parenting was found as a function of positive family processes, which are then predictive of Chinese youths’ psychosocial maturity. Therefore, it is plausible that parenting behavior acts as a crucial mediator to transit the effects of family relationships, the couple’s cohesion, marital intimacy, and other home interpersonal dynamics on youth development [13, 51]." (line 2360-2364)

13.Yeung, J. W. K.; Chan, Y.-C., Parents' Religiosity, Family Socialization and the Mental Health of Children in Hong Kong: Do Raters Make a Difference? Journal of Family Studies 2015, 22, (2), 140-161.
51.Martinez, I.; Murgui, S.; Garcia, O. F.; Garcia, F., Parenting and adolescent adjustment: The mediational role of family self-esteem. Journal of Child and Family Studies 2021, 30, 1184–1197.

For the academic editor

# My only remaining suggestion is that in Section 3.1, we should be told the date (year) when data was collected.

Reply: For the data collection period of the study, it was conducted during Sep to Dec, 2015.

If any, please feel free to let me know, thank you for your attention.
Best regards.

Jerf Yeung